# Clinical Outcomes of Different Calcified Culprit Plaques in Patients with Acute Coronary Syndrome

**DOI:** 10.3390/jcm11144018

**Published:** 2022-07-11

**Authors:** Fangmeng Lei, Yanwei Yin, Xiaohui Liu, Chao Fang, Senqing Jiang, Xueming Xu, Sibo Sun, Xueying Pei, Ruyi Jia, Caiying Tang, Cong Peng, Song Li, Lulu Li, Yini Wang, Huai Yu, Jiannan Dai, Bo Yu

**Affiliations:** 1Department of Cardiology, The 2nd Affiliated Hospital of Harbin Medical University, Harbin 150086, China; lfm1996@163.com (F.L.); dryin22@163.com (Y.Y.); drliuxiaohui1983@163.com (X.L.); fangchaodr@163.com (C.F.); jiangsenqingdr@163.com (S.J.); drxuemingxu@163.com (X.X.); sun15545142240@163.com (S.S.); peixueying0929@163.com (X.P.); hmu_jiaruyi@163.com (R.J.); tangcaiyingtcy@163.com (C.T.); pengcong9710@163.com (C.P.); wsylisong@163.com (S.L.); lulu1039415152@163.com (L.L.); yiniwang8165@163.com (Y.W.); yuhuai1111@163.com (H.Y.); 2Key Laboratory of Myocardial Ischemia, Chinese Ministry of Education, 246 Xuefu Road, Nangang District, Harbin 150086, China

**Keywords:** acute coronary syndrome, calcified plaque, major adverse cardiac events, optical coherence tomography

## Abstract

Background: Previous studies have found that coronary artery calcification is closely associated with the occurrence of major adverse cardiac events (MACE). This study aimed to investigate the characteristics and clinical outcomes of different calcified plaques in patients with acute coronary syndrome (ACS) by using optical coherence tomography (OCT). Methods: 258 ACS patients with calcified culprit plaques who underwent OCT-guided stent implantation were enrolled. They were divided into three subtypes based on the calcified plaque morphology, including eruptive calcified nodules, calcified protrusion, and superficial calcific sheet. Results: Compared with superficial calcific sheet and calcified protrusion, eruptive calcified nodules had the greatest calcium burden and a higher rate of stent edge dissection (*p* < 0.001) and incomplete stent apposition (*p* < 0.001). In a median follow-up period of 2 years, 39 (15.1%) patients experienced MACE (a composite event of cardiac death, target-vessel myocardial infarction, ischemia-driven revascularization), with a significantly higher incidence in the eruptive calcified nodules group (32.1% vs. 10.1% vs. 13.0%, *p* = 0.001). A multivariate Cox analysis demonstrated that the eruptive calcified nodules (hazard ratio 3.14; 95% confidence interval, 1.64–6.02; *p* = 0.001) were an independent predictor of MACE. Conclusions: MACE occurred more frequently in ACS patients with eruptive calcified nodules, and the eruptive calcified nodules were an independent predictor of MACE.

## 1. Introduction

In recent years, studies have found that coronary artery calcification (CAC) is closely associated with the occurrence of major adverse cardiac events (MACE) [1,2]. Moreover, patients with severely calcified lesions had poorer clinical outcomes during and post percutaneous coronary intervention (PCI), including an increased risk of coronary dissection, interventional failure, target lesion revascularization, and long-term mortality [3].

With the development of intravascular imaging, optical coherence tomography (OCT) can identify the microstructures of calcified plaques [4,5] with a resolution of 10–20 µm. Sugiyama et al. [6] identified calcified culprit plaques as three subtypes based on the plaque morphology in patients with acute coronary syndromes (ACS) using OCT, including eruptive calcified nodules, calcified protrusion, and superficial calcific sheet. Subsequently, Nakajima et al. [7] further compared the post-stent OCT findings among these three calcified culprit plaque subtypes. However, the prognosis of three subtypes of calcified culprit plaques is not fully understood. Therefore, the present study aimed to investigate the characteristics and clinical outcomes of ACS patients with different calcified plaques after stent implantation.

## 2. Methods

### 2.1. Study Population

This is a single-center retrospective observational study. From January 2017 to December 2019, a total of 2706 ACS patients underwent OCT imaging of culprit lesions during emergency procedures in the Second Affiliated Hospital of Harbin Medical University. Among them, 314 patients had 314 calcified culprit plaques, and 56 patients were further excluded for the reasons provided in Figure 1. Finally, 258 calcified culprit lesions that were suitable for evaluation before and after stent implantation were included in the final analysis. According to the morphology of the calcified culprit lesions, they were divided into three groups: the eruptive calcified nodules group (*n* = 56); the calcified protrusion group (*n* = 23); and the superficial calcific sheet group (*n* = 179). The clinical data, angiography results, OCT findings before and after stent implantation, and clinical outcomes were compared among the 3 groups. The diagnosis of ACS has been detailed in the Methods of the Appendix A.

The culprit lesion was identified based on angiographic findings, electrocardiogram changes, and/or left ventricular wall motion abnormalities. In patients with multiple stenoses, the plaque with the most severe stenosis or with evidence of acute thrombus on angiography or OCT was considered to be the culprit.

This study was approved by the Ethics Committee of The Second Affiliated Hospital of Harbin Medical University and conducted in accordance with the Declaration of Helsinki. Written informed consent was obtained from all enrolled patients. The data that support the findings of this study are available from the corresponding author upon reasonable request.

### 2.2. Angiographic Analysis and Procedures

A quantitative coronary angiography analysis was performed using the Cardiovascular Angiography Analysis System (CAAS) version 5.10 (Pie Medical Imaging B.V., Maastricht, The Netherlands). All culprit lesion analyses were performed by two independent investigators who were blinded to the clinical data and OCT analysis results. The reference vessel diameter, minimal lumen diameter, diameter stenosis, and lesion length were measured from end-diastolic frames and calibration using the catheter’s tip [8]. Coronary flow was assessed according to the Thrombolysis in Myocardial Infarction (TIMI) Flow Grade [9]. The lesion complexity was assessed using the American College of Cardiology/American Heart Association classification [10]. Calcification was identified as readily apparent radiopacities within the vascular wall at the site of the stenosis [11].

### 2.3. OCT Image Acquisition and Analysis

OCT imaging was performed at the discretion of an interventional cardiologist using a commercially available frequency-domain OCT system (ILUMIEN OPTIS or OPTIS Integrated System, Abbott Vascular, Santa Clara, CA, USA). All OCT images were analyzed in the imaging core laboratory by two experienced investigators who were blinded to the patients’ information. When there was disagreement between the investigators, a consensus reading was performed by a third investigator. For patients with TIMI < 2 and occlusive thrombosis, manual thrombectomy was allowed before OCT imaging. The minimal lumen area was the minimal value of the lumen area along the culprit lesion. The average reference lumen area was defined as the average of the largest lumen area at the proximal and distal ends of the stenosis in the 5 mm segment. The area stenosis percentage referred to the degree of the lumen area of the narrowest frame, and the formula was as follows: [(average reference lumen area-minimal blood flow area)/average reference lumen area] × 100%. Calcified plaque was identified by the presence of superficial substantive calcification at the culprit site without evidence of ruptured lipid plaque [12]. The calcified plaques were divided into three subtypes according to the plaque morphology [6]. In brief, the superficial calcific sheet was defined as a non-protruding calcified plaque (Figure 2A); eruptive calcified nodules as the expulsion of small calcific nodules into the lumen (Figure 2B); and calcified protrusion as a protruding calcific mass without eruptive nodules (Figure 2C). The Methods of the Supplement describe the qualitative, quantitative, and post-procedure analysis of the calcified plaque characteristics, as well as inter- and intra-observer agreement.

### 2.4. Clinical Follow-Up

The patients received scheduled follow-ups at 1, 3, 6, and 12 months and annually thereafter by clinical visit or telephone interview after discharge. The primary endpoints were major adverse cardiac events, which were defined as composite events of cardiac death, target-vessel myocardial infarction (MI), and ischemic-driven revascularization (IDR). The secondary endpoints included non-fatal stroke, major bleeding, and rehospitalization caused by unstable or progressive angina. Detailed definitions of the individual outcome measures have been provided in the Methods of the Supplement. The adverse events were adjudicated by three experienced cardiologists who reviewed the original source documents and were unaware of the baseline OCT data.

### 2.5. Statistical Analysis

The categorical data are presented as counts and percentages and compared using the chi-square test or Fisher exact test, as appropriate. Bonferroni’s correction was applied for multiple comparisons among the three groups and *p* < 0.017 in a two-group comparison was considered significant. Continuous variables are presented as mean ± SD or as median (25th–75th percentiles) for normally and non-normally distributed variables, respectively, and compared with the use of ANOVA or the Kruskal–Wallis H test, and a post-hoc test was used only if *p* < 0.05. Data distribution was assessed according to the Kolmogorov–Smirnov test. The cumulative incidence of survival-free periods from clinical events was estimated using the Kaplan–Meier method. In case of significant differences, pairwise post hoc tests were performed with Bonferroni correction. A multivariate Cox proportional hazards analysis was used to identify the independent predictors of MACE, which included variables *p* < 0.05 in a univariate analysis and factors that may have impacts on MACE. These variables were calcified type, demographic characteristics, risk factors of coronary heart disease, laboratory data, procedural characteristics, and the qualitative and quantitative features of OCT. *p* < 0.05 was regarded as statistically significant. Inter- and intra-observer differences were quantified using the κ coefficient of agreement for the plaque classification. All analyses were performed with SPSS (version 25 for Windows; SPSS, Inc., Chicago, IL, USA).

## 3. Results

A total of 258 calcified culprit plaques were included in the final analysis: 56 (21.7%) were eruptive calcified nodules; 23 (8.9%) were calcified protrusions; and 179 (69.4%) were superficial calcific sheets.

### 3.1. Baseline Clinical Characteristics of Different Calcified Plaques

Clinical characteristics of the three groups are presented in Table 1. Compared with the superficial calcific sheet group, the calcified protrusion and eruptive calcified nodule groups had a higher rate of chronic kidney disease (*p* = 0.001); previous myocardial infarction (*p* < 0.001); and previous PCI (*p* = 0.002).

### 3.2. Angiographic Findings and Procedural Characteristics of Different Calcified Plaques

The coronary angiographic findings and procedural characteristics are shown in Table 2. The angiographic findings showed that most lesions were located in the left anterior descending artery, while 30.4% of the eruptive calcified nodules, 20.7% of the superficial calcific sheets, and 13.0% of the calcified protrusions were located in the right coronary artery (*p* = 0.013). The eruptive calcified nodules group had the largest stent length (*p* = 0.027) and the greatest pressure of pre-dilation (*p* = 0.004) than the other groups. The use of rotational atherectomy (*p* = 0.001) was most frequently observed in eruptive calcified nodules.

### 3.3. Morphological Characteristics of Different Calcified Plaques before and after Stent Implantation

The OCT findings before and after PCI are summarized in Table 3. The eruptive calcified nodules and superficial calcific sheets had a greater reference lumen area (*p* = 0.008) and severe area stenosis (*p* = 0.023) than the calcified protrusions. Calcium burden (*p* < 0.001) was greatest in the eruptive calcified nodules, followed by the superficial calcific sheets and calcified protrusions. Layered plaque was most frequently observed in the superficial calcific sheets (*p* = 0.009). Macrophage accumulation was most frequently observed in the superficial calcific sheets and eruptive calcified nodules (*p* = 0.001). Compared with the superficial calcific sheets and calcified protrusions, the eruptive calcified nodules were more frequently accompanied with thrombus (*p* < 0.001), and the most common type of thrombus was red thrombus.

The post-stent findings showed that the eruptive calcified nodules group had a higher rate of stent edge dissection (*p* < 0.001), incomplete stent apposition (*p* < 0.001), and calcium fractures (*p* < 0.001) than the other groups, and the stent eccentricity was the largest (*p* = 0.003).

### 3.4. Clinical Outcomes

Table 4 shows the clinical outcomes. The clinical follow-up data were available for 258 patients. In a median follow-up period of 2 (1–3) years, the total incidence of MACE was 15.1%. Among the three groups, the percentage of MACE was highest in the eruptive calcified nodules group (32.1% vs. 13.0% vs. 10.1%, *p* = 0.001), mainly from target vessel MI (8.9% vs. 8.7% vs. 1.7%, *p* = 0.016) and IDR (16.1% vs. 13.0% vs. 5.6%, *p* = 0.029). The Kaplan–Meier survival curve analysis is shown in Figure 3.

The predictors of MACE are shown in Table 5. The results of the univariate analysis indicated that eruptive calcified nodules, chronic kidney disease, age, high-sensitivity C-reactive protein, maximal calcification arc, thrombus, and layered plaque were significantly associated with MACE. The multivariate analysis showed that eruptive calcified nodules [hazard ratio (HR): 3.14; 95% confidence interval (CI): 1.64–6.02; *p* = 0.001] and age (HR: 1.05; 95% CI: 1.01–1.09; *p* = 0.009) were independent predictors of MACE.

## 4. Discussion

To the best of our knowledge, this is the first study to compare the clinical outcomes among the three subtypes of calcified plaques in ACS patients with stent implantation. The main findings were as follows: (1) the prevalence of three subtypes were 21.7% for eruptive calcified nodules, 8.9% for calcified protrusion, and 69.4% for superficial calcific sheet; (2) previous myocardial infarction and chronic kidney disease were most frequently observed in the eruptive calcified nodules and calcified protrusions groups; (3) eruptive calcified nodules had the greatest calcium burden and higher macrophage accumulation, followed by superficial calcific sheets and calcified protrusions, and the percentage of stent edge dissection and incomplete stent apposition was highest in the eruptive calcified nodules group; (4) the incidence of MACE was highest in the eruptive calcified nodules group than the other groups, mainly from target-vessel MI and IDR; the eruptive calcified nodules were an independent predictor of MACE.

### 4.1. Classification of Calcified Plaque and Baseline Characteristics of Patients

Previous studies [6] have summarized the diagnosis algorithm of calcified plaques in ACS patients for the first time and divided calcified plaques into three subtypes: eruptive calcified nodules (25.5%); calcified protrusion (7.1%); and superficial calcific sheet (67.4%). The prevalence of the three subtypes in our study was in line with previous observation. Eruptive calcified nodules and calcified protrusions were most frequently observed in ACS patients with previous myocardial infarction and chronic kidney disease. As Lee [13] et al. reported, eruptive calcified nodules are often observed in patients with chronic kidney disease and severe CAC lesions with percutaneous coronary intervention. Calcified protrusion is a protruding calcified mass that is similar to eruptive calcified nodules, so the two may show the same trend of change in some characteristics.

### 4.2. OCT Characteristics of Different Calcified Plaques before and after PCI

In the present study, the eruptive calcified nodules had the greatest calcium burden and higher macrophage accumulation, followed by the superficial calcific sheets and calcified protrusions. These characteristics were consistent with the process of intimal calcification. Inflammatory cells, especially the macrophage phenotype, may induce vascular calcification [14]. Calcification deposits form a calcified plate until the rupture of overlying tissue causes calcification to protrude into the lumen. An OCT study [13] of 889 de novo lesions showed that lesions presenting as calcified nodules had larger calcification angles, longer lengths, greater thickness, shallower locations, and presented more frequently with red thrombus, which was in line with our study, indicating that eruptive calcified nodules had the greatest calcium burden and were prone vulnerability.

It was difficult to allow a balloon or stent to expand directly with calcium plaque because of the hard calcium inside, so pretreatment was required to obtain an acceptable lumen area, especially with severe calcification. In this study, the use of rotational atherectomy was most frequent and the pressure of pre-dilation was greatest in the eruptive calcified nodules, as compared with the superficial calcific sheets and calcified protrusions. However, the methods of pretreatment increase the risk of complications [15], such as coronary artery dissection and coronary artery rupture. The insufficient preparation of these calcified plaques could lead to difficulties in stent delivery, under-expansion, asymmetric expansion, and incomplete stent apposition. The present study reported that the rates of stent edge dissection (73.2%) and incomplete stent apposition (62.5%) were highest in the eruptive calcified nodule group. As Khalifa et al. [16] reported, the frequency of incomplete stent apposition (71%) and stent edge dissection (44%) was highest in the OCT-calcified nodules group, which may be associated with the highest risk of stent restenosis and thrombosis.

### 4.3. Clinical Outcomes and Calcification-Related Predictors

A previous OCT study [17] found that ACS patients with calcified nodules in culprit plaques were more likely to have revascularization within 500 days. An angiographic study [18] also found that patients with moderate or severe CAC were associated with a higher rate of death, myocardial infarction, and IDR. Even the emergence of the new generation of drug-eluting stents was accompanied by a low incidence of events. Similarly, we found that patients with eruptive calcified nodules had the highest incidence of MACE. The reasons for this phenomenon may be that (1) Patients with eruptive calcified nodules had the largest calcium burden and more common usage of pretreatment, which leads to stent edge dissection and incomplete stent apposition. Both together with stent under-expansion were considered to be the causes of in-stent restenosis and in-stent thrombosis [19,20]. (2) The smaller the lumen area after eccentric calcification treatment, the larger the stent eccentricity index, and the more obvious the coronary flow restriction [21]. Moreover, the entry of eccentric protrusions into the lumen causes local blood flow obstruction and high endothelial shear stress, leading to platelet aggregation and thrombosis [22,23]. (3) The present study found that 75.0% of patients with a calcium fracture, which was due to the “hinge movement” [13], had greater torsional stress of the coronary arteries. This may cause the subsequent fracture of the stent and increase the possibility of recurrent ischemia and revascularization. (4) From a pathological point of view, smooth muscle cell apoptosis provides calcium attachment points for calcification, meaning that there are few smooth muscle cells in calcified plaques, and the neointimal coverage may be poor, which leads to in-stent thrombosis.

Based on the above discussion, our conclusion that eruptive calcified nodules and age were independent predictors of MACE was plausible. Calcification in coronary plaques increased with age, especially after the age of 70, regardless of gender [24].

### 4.4. Clinical Significance

This study found that there were significant differences in the clinical outcomes of different calcified culprit plaques after stenting. The eruptive calcified nodules were an independent predictor of MACE, which was consistent with the results shown in previous pathology and follow-up studies [25,26,27]. Calcification was a hallmark of advanced atherosclerosis and it was associated with an overall increase in the number of plaques, a higher risk of future adverse events, and worse outcomes after surgery or percutaneous revascularization, especially the severe CAC at the culprit lesion. These studies indicate that eruptive calcified nodules with a severe calcium burden may be more vulnerable. The current diagnosis and treatment concept is focused on the early identification of vulnerable patients, early prevention, and individualized treatment [28]. Therefore, such patients should be identified in advance and some measures should be taken to prevent the progression and expansion of calcification, for example, the use of statins and stents that are resistant to breakage and constant movement of the hinge.

## 5. Limitations

This study has several limitations. First, this study is a single-center retrospective study, so selection bias is inevitable. Secondly, a propensity score matched analysis was not performed due to the inadequate number of patients, and our study population consisted entirely of culprit lesions. Therefore, our conclusions cannot be generalized, and further large-sample experiments are needed to verify them. Third, OCT may underestimate deep calcifications, thick calcifications, and calcifications behind lipid-rich components due to limited penetration depth. Fourth, performing a balloon dilatation or rotational atherectomy and OCT scanning may cause some surgery-related injuries and affect the visualization of plaque. Fifth, this study did not compare the OCT characteristics of calcified plaques after stenting during follow-up, such as neointimal thickness, in-stent thrombosis, in-stent restenosis, etc. As the study population is limited, we intend to further increase the sample size to observe whether the OCT characteristics of these patients are different during follow-up, in order to better explain the reasons for the clinical outcomes.

## 6. Conclusions

Among these different subtypes of calcified culprit plaques, eruptive calcified nodules had the greatest calcium burden, a higher prevalence of vulnerable features, and were accompanied with more stent edge dissection and incomplete stent apposition after stenting, which may lead to an increased risk of MACE. The eruptive calcified nodules were an independent predictor of MACE.

## Figures and Tables

**Figure 1 jcm-11-04018-f001:**
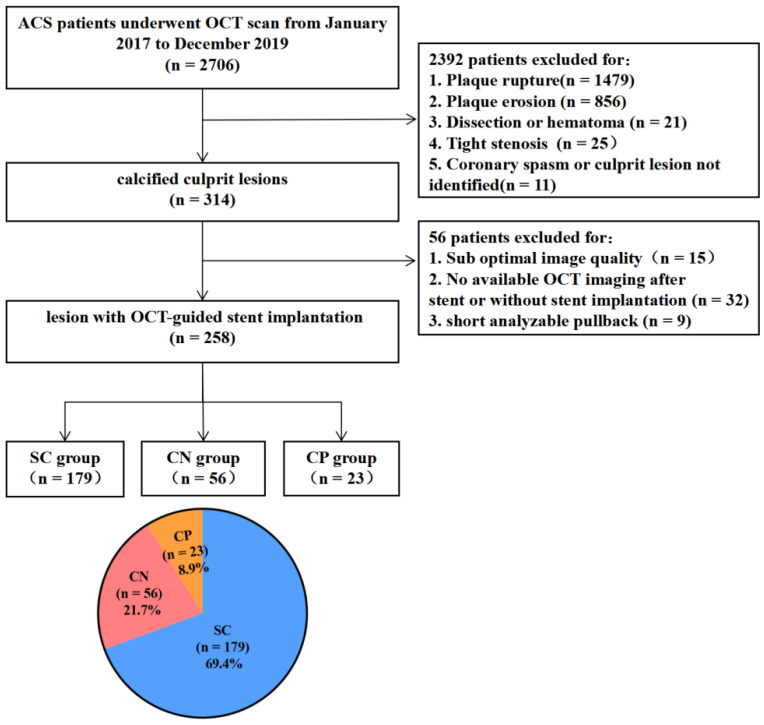
The study flow chart. ACS = acute coronary syndrome; CN = eruptive calcified nodules; CP = calcified protrusion; OCT = optical coherence tomography; SC = superficial calcific sheet.

**Figure 2 jcm-11-04018-f002:**
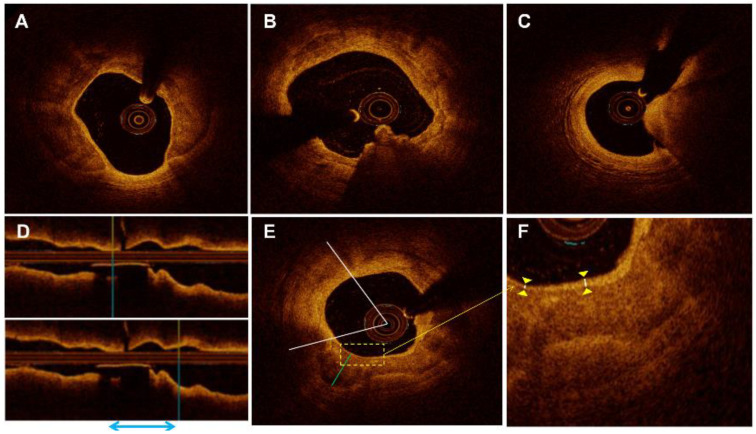
Representative optical coherence tomography images. (**A**) Superficial calcific sheet; (**B**) eruptive calcified nodules; (**C**) calcified protrusion; (**D**,**E**) measurement indicators of calcification. (**D**) The distance between the blue double arrows shows the length of calcification. (**E**) The green line represents the thickness of calcification. The white lines represent the angle of calcification. The yellow box represents the depth of calcification. (**F**) The yellow triangles represent the enlarged image of the depth.

**Figure 3 jcm-11-04018-f003:**
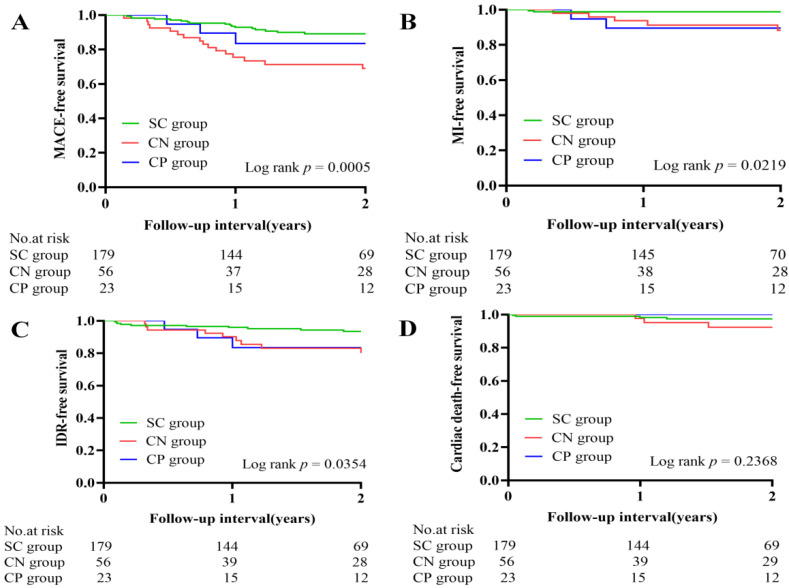
Kaplan–Meier survival curves of clinical outcomes. (**A**) MACE-free survival; (**B**) target vessel myocardial infarction-free survival; (**C**) ischemia-driven revascularization-free survival; (**D**) cardiac death-free survival. MACE, major adverse cardiac events; MI, myocardial infarction-free survival; IDR, ischemia-driven revascularization.

**Table 1 jcm-11-04018-t001:** Baseline clinical characteristics of different calcified culprit plaques.

Variables	SC Group (*n* = 179)	CN Group (*n* = 56)	CP Group (*n* = 23)	*p* Value	*p* * Value		
				SC vs. CN vs. CP	SC vs. CN	SC vs. CP	CN vs. CP
Age, years	65.0 ± 9.3	65.9 ± 9.5	65.2 ± 7.9	0.809	NA	NA	NA
**Sex**				0.995	NA	NA	NA
Male, *n* (%)	117 (65.4)	37 (66.1)	15 (65.2)				
Female, *n* (%)	62 (34.6)	19 (33.9)	8 (34.8)				
**Clinical presentation**				0.919	NA	NA	NA
STEMI, *n* (%)	89 (49.7)	25 (44.6)	10 (43.5)				
NSTEMI, *n* (%)	21 (11.7)	6 (10.7)	3 (13.0)				
UAP, *n* (%)	69 (38.5)	25 (44.6)	10 (43.5)				
**Risk factors**							
Hypertension, *n* (%)	103 (57.5)	42 (75.0)	14 (60.9)	0.064	NA	NA	NA
Diabetes mellitus, *n* (%)	43 (24.0)	22 (39.3)	6 (26.1)	0.082	NA	NA	NA
Dyslipidemia, *n* (%)	103 (57.5)	30 (53.6)	14 (60.9)	0.806	NA	NA	NA
**Smoking status**				0.139	NA	NA	NA
Never	107 (59.8)	30 (53.6)	12 (52.2)				
Former	16 (8.9)	12 (21.4)	2 (8.7)				
Current	56 (31.3)	14 (25.0)	9 (39.1)				
**Clinical history**							
CKD, *n* (%)	1 (0.6)	5 (8.9)	2 (8.7)	0.001	0.003	0.035	1.000
Previous MI, *n* (%)	15 (8.4)	11 (19.6)	9 (39.1)	<0.001	0.036	<0.001	0.127
Previous PCI, *n* (%)	17 (9.5)	10 (17.9)	8 (34.8)	0.002	0.141	0.002	0.182
Previous CABG, *n* (%)	1 (0.6)	1 (1.8)	0 (0)	0.519	NA	NA	NA
**Laboratory data**							
TC, mg/dL	168.9 ± 48.7	169.5 ± 44.9	152.3 ± 39.9	0.286	NA	NA	NA
TG, mg/dL	117.0 (85.1, 169.2)	137.3 (88.2, 209.1)	136.4 (98.8, 154.6)	0.486	NA	NA	NA
LDL-C, mg/dL	101.7 ± 39.1	99.9 ± 37.9	83.4 ± 29.9	0.108	NA	NA	NA
HDL-C, mg/dL	49.7 ± 16.9	46.4 ± 12.9	45.3 ± 11.8	0.243	NA	NA	NA
CK-MB, ug/L	2.3 (0.8, 20.7)	1.8 (0.6, 7.2)	1.2 (0.5, 22.2)	0.253	NA	NA	NA
cTnI, ug/L	0.30 (0.02, 5.24)	0.15 (0.02, 1.52)	0.28 (0.02, 3.04)	0.290	NA	NA	NA
HbA1c, %	6.3 ± 1.5	6.4 ± 1.2	6.2 ± 1.3	0.782	NA	NA	NA
hs-CRP, mg/L	3.2 (1.2, 9.4)	2.5 (1.0, 5.9)	2.5 (0.9, 10.8)	0.744	NA	NA	NA
LVEF, %	58.2 ± 7.2	57.7 ± 7.8	57.3 ± 7.8	0.834	0.661	0.633	0.872
**Medication at discharge**							
Aspirin, *n* (%)	177 (98.9)	56 (100)	23 (100)	0.641	NA	NA	NA
Clopidogrel/Tigrillo, *n* (%)	178 (99.4)	55 (98.2)	23 (100)	0.597	NA	NA	NA
Statin, *n* (%)	176 (98.3)	56 (100)	23 (100)	0.512	NA	NA	NA
ACEI/ARB, *n* (%)	89 (49.7)	31 (55.4)	15 (65.2)	0.329	NA	NA	NA
β–blockers, *n* (%)	113 (63.1)	36 (64.3)	15 (65.2)	0.973	NA	NA	NA
CCB, *n* (%)	27 (15.1)	11 (19.6)	3 (13.0)	0.705	NA	NA	NA

Values expressed as mean ± standard deviation, median (25th, 75th percentiles), or *n* (%). A *p*-value < 0.05 or *p* *-value < 0.017 was considered statistically significant. ACE-I, angiotensin-converting enzyme inhibitor; ARB, angiotensin II receptor blocker; CABG, coronary artery bypass grafting; CCB, calcium channel blockers; CKD, chronic kidney disease; CK-MB, creatine kinase-MB; CN, eruptive calcified nodules; CP, calcified protrusion; cTnI, cardiac troponin I; HbA1c, glycated hemoglobin A1c; HDL-C, high-density lipoprotein cholesterol; hs-CRP, high-sensitivity C-reactive protein; LDL-C, low-density lipoprotein cholesterol; LVEF, left ventricle ejection fraction; MI, myocardial infarction; NA, non-available; NSTEMI, non-ST-elevation myocardial infarction; PCI percutaneous coronary intervention; SC, superficial calcific sheet; STEMI, ST-elevation myocardial infarction; TC, total cholesterol; TG, total triglyceride; UAP, unstable angina pectoris.

**Table 2 jcm-11-04018-t002:** Angiographic findings and procedural characteristics of different calcified culprit plaques.

Variables	SC Group (*n* = 179)	CN Group (*n* = 56)	CP Group (*n* = 23)	*p* Value	*p* * Value		
				SC vs. CN vs. CP	SC vs. CN	SC vs. CP	CN vs. CP
Culprit vessel				0.013	0.130	0.009	0.093
Left anterior descending artery	128 (71.5)	32 (57.1)	13 (56.5)				
Left circumflex artery	14 (7.8)	7 (12.5)	7 (30.4)				
Right coronary artery	37 (20.7)	17 (30.4)	3 (13.0)				
TIMI flow grade 0–1	56 (31.3)	12 (21.4)	7 (30.4)	0.362	NA	NA	NA
Multivessel disease	156 (87.2)	54 (96.4)	22 (95.7)	0.081	NA	NA	NA
**Quantitative coronary angiography analysis**							
Lesion length, mm	31.3 ± 10.7	33.4 ± 12.6	28.6 ± 10.5	0.186	NA	NA	NA
Minimal lumen diameter, mm	0.65 ± 0.32	0.63 ± 0.33	0.60 ± 0.29	0.766	NA	NA	NA
Reference vessel diameter, mm	3.12 ± 0.62	3.18 ± 0.62	3.06 ± 0.64	0.670	NA	NA	NA
Diameter stenosis, %	78.7 ± 10.7	79.6 ± 10.8	79.4 ±9.2	0.844	NA	NA	NA
**Procedural characteristics**							
Number of stents, *n*	1.3 ± 0.5	1.5 ± 0.6	1.3 ± 0.4	0.072	NA	NA	NA
Multiple stents, *n* (%)	52 (29.1)	25 (44.6)	6 (26.1)	0.075	NA	NA	NA
Stent length, mm	35.9 ± 12.3	40.7 ± 13.2	34.0 ± 14.3	0.027	0.014	0.496	0.034
Stent diameter, mm	3.14 ± 0.36	3.16 ± 0.36	3.05 ± 0.49	0.673	NA	NA	NA
Rotational atherectomy, *n* (%)	9 (5.0)	12 (21.4)	1 (4.3)	0.001	<0.001	1.000	0.094
Thrombectomy, *n* (%)	7 (3.9)	3 (5.4)	0 (0)	0.764	NA	NA	NA
Pre-dilation, *n* (%)	173 (96.6)	53 (94.6)	22 (95.7)	0.574	NA	NA	NA
Post-dilation, *n* (%)	165 (92.2)	51 (91.1)	22 (95.7)	0.867	NA	NA	NA
Pressure of pre-dilation, atm	13.4 ± 3.7	15.5 ± 3.3	13.8 ± 4.3	0.004	0.001	0.742	0.101
Pressure of post-dilation, atm	20.5 ± 3.6	20.2 ± 3.3	20.4 ± 3.0	0.929	NA	NA	NA
Thrombus aspiration	68 (38.0)	18 (32.1)	7 (30.4)	0.613	NA	NA	NA

Values expressed as mean ± standard deviation, median (25th, 75th percentiles), or *n* (%). A *p*-value < 0.05 or *p* *-value < 0.017 was considered statistically significant. CN, eruptive calcified nodules; CP, calcified protrusion; NA, non-available; SC, superficial calcific sheet.

**Table 3 jcm-11-04018-t003:** Optical coherence tomography analysis of different calcified culprit plaques.

Variables	SC Group (*n* = 179)	CN Group (*n* = 56)	CP Group (*n* = 23)	*p* Value	*P* * Value		
				SC vs. CN vs. CP	SC vs. CN	SC vs. CP	CN vs. CP
**Preprocedural optical coherence tomography analysis**							
Reference lumen area, mm^2^	6.32 ± 2.06	6.70 ± 2.92	4.96 ±1.91	0.008	0.278	0.007	0.002
Minimal lumen area, mm^2^	1.45 ± 0.53	1.63 ± 0.71	1.58 ± 0.74	0.117	0.049	0.338	0.717
Area stenosis, %	76.8 ± 10.2	75.7 ± 8.9	70.5 ± 14.1	0.023	0.483	0.006	0.043
Calcification length, mm	15.5 ± 7.3	22.2 ± 9.5	9.9 ± 4.3	<0.001	<0.001	0.001	<0.001
Mean calcification arc, °	149.6 ± 37.7	194.4 ± 44.2	90.3 ± 19.4	<0.001	<0.001	<0.001	<0.001
Maximal calcification arc, °	249.8 ± 70.6	320.9 ± 50.0	137.3 ± 44.1	<0.001	<0.001	<0.001	<0.001
Mean calcification depth, μm	90.0 (60.0, 130.0)	50.0 (30.0, 70.0)	120.0 (87.5, 170.0)	<0.001	<0.001	0.007	<0.001
Minimal calcification depth, μm	10.0 (10.0, 20.0)	0.0 (0.0, 10.0)	30.0 (20.0, 50.0)	<0.001	<0.001	0.002	<0.001
Mean calcification thickness, μm	758.1 ± 156.3	789.3 ± 146.2	854.4 ± 207.3	0.018	0.202	0.007	0.100
Maximal calcification thickness, μm	1134.5 ± 241.2	1242.7 ± 257.3	1198.7 ± 251.6	0.013	0.004	0.239	0.470
Calcification index	2002.2 (1402.5, 2981.9)	3888.6 (2963.6, 5561.7)	878.0 (615.1, 1187.0)	<0.001	<0.001	<0.001	<0.001
Layered plaque, *n* (%)	141 (78.8)	35 (62.5)	13 (56.5)	0.009	0.023	0.036	0.810
Macrophage, *n* (%)	162 (90.5)	55 (98.2)	16 (69.6)	0.001	0.081	0.010	0.001
Microchannel, *n* (%)	36 (20.1)	7 (12.5)	3 (13.0)	0.376	0.277	0.578	1.000
Cholesterol crystal, *n* (%)	51 (28.5)	16 (28.6)	8 (34.8)	0.819	1.000	0.703	0.782
Thrombus	94 (52.5)	54 (96.4)	11 (47.8)	<0.001	<0.001	0.840	<0.001
Red	12 (6.7)	31 (55.4)	4 (17.4)				
White	73 (40.8)	15 (26.8)	7 (30.4)				
Mixed	9 (5.0)	8 (14.3)	0 (0)				
**Postprocedural optical coherence tomography analysis**							
Reference lumen area, mm^2^	7.91 ± 2.05	8.23 ± 2.83	6.99 ± 1.87	0.081	0.352	0.063	0.025
Minimal stent area, mm^2^	4.26 ± 1.41	4.46 ± 1.53	4.40 ± 1.88	0.656	0.385	0.665	0.880
Stent edge dissection, *n* (%)	77 (43.0)	41 (73.2)	7 (30.4)	<0.001	<0.001	0.354	0.001
Proximal edge dissection, *n* (%)	6 (3.4)	2 (3.6)	0 (0)	1.000	1.000	1.000	1.000
Distal edge dissection, *n* (%)	8 (4.5)	1 (1.8)	0 (0)	0.635	0.690	0.601	1.000
In-stent dissection, *n* (%)	67 (37.4)	39 (69.6)	7 (30.4)	<0.001	<0.001	0.670	0.003
ISA, *n* (%)	56 (31.3)	35 (62.5)	8 (34.8)	<0.001	<0.001	0.919	0.046
Maximal ISA distance, *μ*m	0.37 ± 0.15	0.39 ± 0.19	0.40 ± 0.13	0.905	0.747	0.712	0.858
In-stent tissue protrusion, *n* (%)	133 (74.3)	40 (71.4)	19 (82.6)	0.609	0.801	0.540	0.451
Smooth protrusion, *n* (%)	55 (30.7)	12 (21.4)	10 (43.5)	0.135	0.240	0.320	0.087
Disrupted fibrous tissue protrusion, *n* (%)	33 (18.4)	11 (19.6)	5 (21.7)	0.921	0.995	0.922	1.000
Irregular protrusion, *n* (%)	52 (29.1)	21 (37.5)	4 (17.4)	0.202	0.304	0.325	0.111
Thrombus	66 (36.9)	21 (37.5)	6 (26.1)	0.579	1.000	0.432	0.477
White	20 (11.2)	9 (16.1)	3 (13.0)				
Red	39 (21.8)	10 (17.9)	3 (13.0)				
Mixed	7 (3.9)	2 (3.6)	0 (0)				
Stent expansion ratio	0.547 ± 0.137	0.554 ± 0.122	0.620 ± 0.147	0.051	0.712	0.015	0.051
Stent under-expansion, *n* (%)	172 (96.1)	55 (98.2)	21 (91.3)	0.301	0.684	0.273	0.202
Stent eccentricity	0.346 ± 0.117	0.402 ± 0.104	0.333 ± 0.108	0.003	0.001	0.624	0.015
Calcium fracture, *n* (%)	77 (43.0)	42 (75.0)	1 (4.3)	<0.001	<0.001	<0.001	<0.001

Values expressed as mean ± standard deviation, median (25th, 75th percentiles), or *n* (%). A *p*-value < 0.05 or *p* *-value < 0.017 was considered statistically significant. CN, eruptive calcified nodules; CP, calcified protrusion; ISA, incomplete stent apposition; SC, superficial calcific sheet.

**Table 4 jcm-11-04018-t004:** Clinical outcomes.

Variables	SC Group (*n* = 179)	CN Group (*n* = 56)	CP Group (*n* = 23)	*p* Value	*p* * Value		
				SC vs. CN vs. CP	SC vs. CN	SC vs. CP	CN vs. CP
MACE	18 (10.1)	18 (32.1)	3 (13.0)	0.001	0.007	0.691	0.234
Cardiac death	5 (2.8)	4 (7.1)	0 (0.0)	0.265	0.171	0.427	0.229
TVMI	3 (1.7)	5 (8.9)	2 (8.7)	0.016	0.011	0.033	0.985
IDR	10 (5.6)	9 (16.1)	3 (13.0)	0.029	0.012	0.136	0.776
Stroke	7 (3.9)	3 (5.4)	1 (4.3)	0.884	0.647	0.854	0.920
Rehospitalization	23 (12.8)	13 (23.2)	4 (17.4)	0.157	0.075	0.502	0.644
Bleeding	3 (1.7)	0 (0)	1 (4.3)	0.343	0.347	0.349	0.131

Values expressed as mean ± standard deviation, median (25th, 75th percentiles), or *n* (%). A *p*-value < 0.05 or *p* *-value < 0.017 was considered statistically significant. CN, eruptive calcified nodules; CP, calcified protrusion; IDR, ischemia-driven revascularization; MACE, major adverse cardiac event; SC, superficial calcific sheet; TVMI, target vessel myocardial infarction.

**Table 5 jcm-11-04018-t005:** Univariate and multivariate Cox proportional hazards model for predictors of MACE.

	Univariate Analysis		Multivariable Analysis	
	HR (95% CI)	*p* Value	HR (95% CI)	*p* Value
Eruptive calcified nodules	3.26 (1.74, 6.13)	<0.001	3.14(1.64, 6.02)	0.001
Sex	1.46 (0.78, 2.76)	0.238		
Age	1.05 (1.02, 1.09)	0.004	1.05(1.01, 1.09)	0.009
Hypertension	1.39 (0.71, 2.75)	0.341		
Diabetes mellitus	1.58 (0.82, 3.05)	0.170		
Dyslipidemia	0.89 (0.47, 1.67)	0.720		
Chronic kidney disease	5.23 (1.84, 14.82)	0.002		
Previous MI	1.13 (0.47, 2.70)	0.781		
Previous PCI	1.01 (0.39, 2.58)	0.986		
LDL-C	1.01 (0.99, 1.01)	0.187		
hs-CRP	1.06 (1.00, 1.13)	0.049		
LVEF	1.00 (0.95, 1.04)	0.915		
Stent length	1.00 (0.97, 1.02)	0.744		
Rotational atherectomy	0.95 (0.29, 3.08)	0.931		
Pressure of predilation	1.06 (0.97, 1.16)	0.166		
Area stenosis	1.00 (0.97, 1.03)	0.844		
Calcification length	1.01 (0.98, 1.05)	0.477		
Mean calcification arc	1.00 (0.99, 1.01)	0.110		
Maximal calcification arc	1.01 (1.00, 1.01)	0.017		
Mean calcification depth	0.99 (0.99, 1.00)	0.069		
Minimal calcification depth	1.00 (0.98, 1.01)	0.674		
Mean calcification thickness	1.00 (0.99, 1.00)	0.443		
Maximal calcification thickness	1.00 (0.99, 1.00)	0.488		
Thrombus	3.10 (1.37, 7.03)	0.007		
Macrophage	2.29 (0.55, 9.56)	0.256		
Layered plaque	2.41 (1.28, 4.54)	0.006		
Stent eccentricity	4.27 (0.30, 60.92)	0.284		
Stent edge dissection	1.13 (0.60, 2.11)	0.710		
Incomplete stent apposition	1.24 (0.64, 2.42)	0.521		
Calcium fracture	1.43 (0.76, 2.70)	0.266		

CI, confidence interval; HR, hazard ratio; hs-CRP, high-sensitivity C-reactive protein; LDL-C, low-density lipoprotein cholesterol; LVEF, left ventricle ejection fraction; MI, myocardial infarction; PCI percutaneous coronary intervention.

## Data Availability

The data that support the findings of this study are available from the corresponding author upon reasonable request.

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
