# Peer review of "Clinical Outcomes of Different Calcified Culprit Plaques in Patients with Acute Coronary Syndrome"

_jcm, 2022, doi:10.3390/jcm11144018_

Round 1
Reviewer 1 Report
This is an attractive study aimed at confirming the value of coronary calcification as categorized by Sugiyama et al. in 2019 to predict survival after ACS using a small single-centre retrospective series of patients undergoing PCI. The authors are to be congratulated for this initiative and a well reported series. Besides a revision of English language by a native reviser (only a few minor changes are needed), some methodological improvements and clarifications are required to support authors' conclusions.
MINOR CHANGES
Overall (including abstract)
- Using “long-term” to describe your outcomes can be misleading when based on a 2-year median follow-up after ACS. Either avoid mentioning a “long-term” duration or use mid-term or mid/long-term instead
-When describing study methods, results, conclusions....... you should always refer to culprit lesions, not to a wider population of coronary patients. You rightly admit this in Limitations but should make it also clear in your statements along the paper.
- Use µm instead of um
Introduction
- Line 1. You mention studies (in plural) but include only a single reference. Please add direct original references supporting your statement
- Lines 2-3. I agree with your point, but this is not obvious; you should mention existing references to support this.
Methods
- Clinical follow-up. "...by hospital visit or phone call...". Please specify: Was this performed prospectively and with a time planning? At what time intervals? Was the phone interviewer qualified (MD?)?
- Statistical analysis. A propensity score matched analysis appears to be an optimal approach to analyse your data according to current standards. Please justify why you are using just multivariate regression instead (maybe inadequate number of patients? Limitations for matching? If so, acknowledge this as a limitation or alternatively perform a propensity score analysis.
- Statistical analysis. “...multivariate ...included variables with p < 0.05..." Despite this being specified in the Supplement, you should report in the main article all the variables (or at least variable types) that were considered for selection for this analysis. This is key information.
Results
- Clinical outcomes. You compare the total incidence of MACE. However, since follow-up durations are not exactly the same for the 3 groups, exposure times may not be comparable. You should use a landmark time for this analysis. For instance: MACE at 1 year or at 2 years (provided enough patients are available). Otherwise, you should only report Kaplan-Meier analysis to allow considering data for the whole varied follow-up durations.
- Kaplan-Meier analysis. Please confirm here that all required baseline and procedural variables were considered for univariate analyses and potential selection for multivariate analysis.
- Kaplan-Meier analysis. For your main conclusions on the group with eruptive calcified nodules, it is not clear to the reader which is the reference group for comparison: other calcification groups (pooled)? calcified protrusion? This is difficult to interpret taking into account you include superficial calcification as an additional variable for univariate analyses.
Discussion
- Please adjust your comments to include changes suggested for previous sections.
Reviewer 2 Report
1. There is a small number in calcified nodule and protrusion lesion. so, I don't know for statistically significant difference among calcified 3 types.
2. describe for LV systolic function. ex) ejection fraction in LV
3. MACE was defined as a composite event of cardiac death, non-fatal myocardial infarction, ischemia-driven revascularization (IDR), non-fatal stroke, major bleeding, and rehospitalization caused by unstable or progressive angina. However, if only OCT target blood vessels are evaluated, MACE should be done as cardiac death, target-vessel MI, and ischemic-driven revascularization. I think we should look at the results with this and look at the rest as 2ndary outcoms.
4. Since the left ventricular EF affects the patient's prognosis, we should check the statistics again by putting the left ventricular EF as a variable.
